# Height–Area–Storage Functional Models for Evaporation-Loss Inclusion in Reservoir-Planning Analysis

**Adebayo J. Adeloye [1],\* , Ibrahim Y. Wuni [2] , Quan V. Dau [1] , B.-S. Soundharajan [3] and K. S. Kasiviswanathan [4]**

1   Institute for Infrastructure and Environment, Heriot-Watt University, Edinburgh EH14 4AS, UK
2   Department of Building and Real Estate, Faculty of Construction and Environment, The Hong Kong Polytechnic University, Hung Hom, Kowloon ZS725, Hong Kong
3   Department of Civil Engineering, Amrita School of Engineering, Coimbatore 641 112, Amrita Vishwa Vidyapeetham, India
4   Department of Water Resources Development and Management, Indian Institute of Technology Roorkee, Roorkee 247 667, India
\*   Correspondence: a.j.adeloye@hw.ac.uk

**Abstract:** Reservoir planning without the explicit accommodation of evaporation loss leads to errors in capacity estimates. However, whenever evaporation loss is considered, its quantification uses linear approximations of the intrinsically nonlinear height–area–storage (H–A–S) relationship to estimate the reservoir area, leading to bias in capacity estimates. In this work, biases resulting from using various H–A–S models are evaluated. These models include linear and nonlinear functions, either specifically developed for the case-study sites or available in the Global Reservoir and Dam (GRanD) database. All empirically derived approximations used data for two dams in India: the Bhakra on Sutlej River and the Pong on the Beas River, both tributaries of the Indus River. The results showed that linear H–A–S models underestimate the exposed surface area of the Pong reservoir by up to 11.19%; the bias at Bhakra was much less. The GRanD H–A–S model performed very poorly at both reservoirs, producing overprediction in exposed reservoir area of up to 100% and 415% at the Pong and Bhakra reservoirs, respectively. Analyses also showed that up to 29% increase in reservoir capacity is required to compensate for the effect of net evaporation loss at low demand levels. As demand increases, the required evaporation-correction capacity decreases in proportional terms and is indistinguishable for all H–A–S models. Finally, recommendations are made on using the results for evaporation adjustment at nongauged sites in the region.

**Keywords:** evaporation loss; storage capacity; reservoir; WEAP; SPA; height–area–storage functions

## 1. Introduction

In arid and semiarid regions where the intermittent nature of river flows creates challenges in meeting water demands, these reservoirs constitute a significant component of water-supply infrastructure [1]. Thus, reservoirs are relied upon to balance the variability in flow profiles associated with droughts or erratic rainfall patterns in order to meet water demand with some degree of reliability [1–3].

Traditionally, planning for sizing reservoirs uses historic runoff data records at the reservoir site and must account for all consumptive demands placed on the system. Some of these demands (e.g., water supply and irrigation) are tangible, and can be readily identified and quantified. However, there are additional water demands that are less tangible but may be significant, in that, if their effects are

omitted, the resulting reservoir size would be based on a wrong or biased measurement [4,5]. Among these less-tangible demands, prominent is evaporation loss [4], although as shown by, e.g., de Araujo et al. [6], capacity reduction due to sedimentation is also important; however, consideration of this is outside the scope of the current study. Thus, as noted by Montaseri and Adeloye [7], it is important to explicitly accommodate the evaporation loss in question in reservoir-planning analysis to avoid mis-sizing the reservoir capacity.

However, a persisting challenge in reservoir-planning and -operation analysis has to do with how to explicitly accommodate evaporation loss in reservoir mass-balance equations that form the basis of such analyses [1,2]. This is founded on the premise that volumetric evaporation loss is determined as the product of net evaporation depth (evaporation rate minus depth of direct precipitation) and the corresponding exposed surface area of the reservoir [8]. Planning analysts have attempted to address this by deploying reservoir height–area–storage (H–A–S) functional formulations to estimate the exposed surface area of reservoirs at different storage states [1,8,9].

Where reservoir planning is based on the versatile behavior-analysis (BA) approach [8,10], H–A–S, or more specifically the area–storage function, must be linear to be tractable. To achieve this, the intrinsically nonlinear area–storage relationship is often approximated by a linear function between the top of the dead storage and the top of the active storage level in the reservoir [9]. However, there has been no systematic scrutiny of the biases that this might produce in the estimated reservoir area, and the effect of biases on capacity.

Possibilities for remedying the bias of linear approximations include using the H–A–S function in its true nonlinear form. Examples of such nonlinear functions are the H–A–S relationships developed from the Global Reservoir and Dam (GRanD) database [11] that are used in many regional and global water-resource studies. The GRanD models were developed using data from several world reservoirs and are nonlinear in conformity with the intrinsic nonlinear nature of the relationships. Another possibility is to use evaporation as water depth as in, e.g., the Water Evaluation and Planning (WEAP) tool [12], which completely removes the need for the exposed surface area.

The aim of this study is to assess the effect of different empirical formulations of H–A–S reservoir relationships on reservoir area and capacity estimates. This was achieved by:

(i). empirically fitting linear and nonlinear H–A–S functions to the observed bathymetric area, volume, and height data for the selected reservoirs, as well as extracting existing functions in the GRanD database;

(ii). assessing the biases or errors associated with the use of various functions for predicting the exposed surface areas of reservoirs at different reservoir storage states;

(iii). carrying out reservoir-planning analyses with and without evaporation consideration, and hence assessing the evaporation effects on capacity estimates for various H–A–S functions; and

(iv). critically examining the results in (ii) and (iii) to identify the most effective model(s) for explicitly accommodating evaporation loss in reservoir-planning analysis and make recommendations.

## 2. Materials and Methods

### 2.1. Data Collection

The study required runoff, evaporation, and rainfall data of case-study reservoirs for the implementation of reservoir-planning analysis. Additionally, topographical data in the form of the corresponding H–A–S relationship at the analyzed reservoir sites were also required for developing the empirical functions to directly incorporate the evaporation loss in planning analysis. The study used the data for two reservoirs located on the Beas and Sutlej Rivers, respectively, the main tributaries of the Indus River in India. The two reservoirs are close, both located in the state of Himachal Pradesh, and are the main case-study reservoirs for an ongoing large-scale research program on sustaining water resources in the Indian Himalayas under climate change (SusHi-Wat: Sustainable Himalayan Water Resources in a Changing Climate, Project NE/N016394/1). The current work contributes to the

overall successful implementation of the research project. All the required data were sourced from the Bhakra-Beas Management Board (BBMB), which is responsible for the management and operation of the two reservoirs.

For the Bhakra reservoir on Sutlej River, the monthly time series of evaporation rates, direct rainfall, inflow, and releases spanning from January 2000 to December 2006 were collected. For the Pong reservoir on Beas River, the monthly data spanned from January 2001 to December 2010. The time series of the net evaporation depth were calculated as the difference between the time series of the measured evaporation rates and direct precipitation for each reservoir. Although these data are relatively short to assess long-term behavior, they were the only ones available from the BBMB at the time. The shortness also informed the decision to employ a monthly rather than annual temporal scale for planning analyses, thereby capturing both the within-year and over-year storage needs. Daily temporal scale is detailed for planning analyses too.

*2.2. Reservoir-Planning Techniques for Accommodating Evaporation Loss*

Since the linear mass-balance equation of a BA is incompatible with a nonlinear H–A–S relationship, alternative reservoir-planning analysis techniques are required when adopting nonlinear H–A–S formulations. Modified sequent peak algorithm mSPA [10] was used in the study because it can handle any H–A–S formulation, whether linear or nonlinear. Apart from this versatility of the mSPA, the approach also has other advantages over a BA, including the uniqueness of its outcome and its immunity against the misbehavior first identified with the BA by Pretto et al. [13], in which the plot against the record length of the median and other statistics of the distribution of capacity exhibited an unusual hump.

All empirical H–A–S models developed as part of the current work and the GRanD model are the area–storage type, while the WEAP tool uses a height–storage formulation. The ways the two formulations are used in the mSPA are described in the following subsections.

### 2.2.1. MSPA with Area–Storage Function

The mSPA involves two main steps: Step 1, in which capacity without accounting for evaporation is estimated, and Step, 2 in which evaporation is included.

Step 1: Approximate (without evaporation) capacity estimation

Let: $K_t$ = cumulative volumetric deficit at the beginning of time $t$, ($m^3$); $K_{t+1}$ = corresponding volumetric deficit at the end of time $t$ or at the beginning of time $t + 1$, ($m^3$); $D_t$ = demand during time $t$, ($m^3$); $Q_t$ = inflow during time $t$, ($m^3$); and N = total number of simulation period. Then, for an initially full reservoir in which there is no deficit, i.e., $K_0 = 0.0$:

(i). Determine $K_{t+1} = \max (0.0, K_t + D_t − Q_t)$ for all time periods, $t = 1, 2, \ldots , N$

(ii). If $K_N = K_0$, then go to (iii); else, if this is the first iteration, set $K_0 = K_N$ and go to (i); else, STOP: the SPA has failed because gross period demand is higher than the average inflow.

(iii). Reservoir active storage capacity, $K_a = \max(K_t)$.

Step 2: Adjustment for volumetric evaporation loss

(i). Determine reservoir states $S_t$ using the $K_a$ and $K_t$ obtained in Step 1, i.e.,

$$S_t = K_a − K_t; 0 \leq t \leq N \tag{1}$$

Note that as a critical period technique, the initial storage state, $S_o = K_a$, i.e., the reservoir is initially full.

(ii). Using $S_t$, determine = corresponding exposed area $A_t$ from the H–A–S model. The mean exposed surface area in interval $[t, t + 1]$ becomes:

$$A_v = 0.5 \, (A_t + A_{t+1}) \tag{2}$$

(iii). Determine the net evaporation volume ($m^3$) in the interval, $EV_t$, as:

$$EV_t = A_v \, (E_t - P_t)$$

where $E_t$ and $P_t$ are the evaporation and precipitation depths (m), respectively, during $t$.

(iv). Rerun Step 1 to now include evaporation. This effectively involves modifying Step 1i to:

$$K_{t+1} = \max \, (0.0, K_t + D_t + EV_t - Q_t); \, 0 \le t \le N$$

Evaporation-adjusted capacity $K_a{}^*$ then becomes:

$$K_a{}^* = \max \, (K_t + 1).$$

(v). However, as noted by [14], the difference between $K_a$ and $K_a{}^*$ may not be entirely due to the inclusion of $EV_t$, but also due to a shift in the critical period. To remove this effect, Montaseri and Adeloye [15] recommend the following iterative steps to obtain the correct evaporation-impacted active storage-capacity estimate:

    a.    Using the estimated $K_a$ and $K_a{}^*$, determine the $\beta = \left| \frac{K_a{}^* - K_a}{K_a} \right|$; if $\beta \le 0.0001$, then STOP, because $K_a{}^*$ is the exact active storage capacity, otherwise, go to step (b)

    b.    Set $K_a = K_a{}^*$

    c.    Determine the new storages ($S_t$) using Equation (1) for all $t = 1, 2, \ldots, N$.

    d.    Determine new storage capacity $K_a{}^*$ by including the $EV_t$ values [8].

    e.    Go to (a) and check the value of $\beta$.

### 2.2.2. MSPA with Height–Storage Function

The WEAP procedure [12] uses evaporation as water depth, and this is also possible with the mSPA. In this case, Step 1 is exactly as described in Section 2.2.1. Step 2 is slightly different and is implemented thus:

(i). Determine $S_t$ ($t = 1, 2, \ldots, N$) using Equation (1). Using $S_t$, determine corresponding height $H_t$ using the height–storage function.

(ii). Adjust the reservoir level for the effect of net evaporation by algebraically deducting the net evaporation depth:

$$H_{t,adj} = H_t - (E_t - P_t) \tag{3}$$

where $H_{t,adj}$ is the adjusted reservoir level (m); $H_t$ is the unadjusted reservoir level (m).

(iii). Convert $H_{t,adj}$ back to adjusted storage $S_{t,adj}$ using the height–storage function.

(iv). Determine the volumetric evaporation loss as:

$$EV_t = S_{t,adj} - S_t$$

(v). Rerun Step 1 to now include evaporation by modifying the expression for $K_{t+1}$ to:

$$K_{t+1} = \max \, (0.0, K_t + D_t + EV_t - Q_t); \, 0 \le t \le N$$

(vi). Complete the necessary checks as described in Section 2.2.1 (Step 2v) to determine evaporation adjusted capacity $K_a$*.

### 2.3. Specification and Parameterization of H–A–S Models

#### 2.3.1. Nonlinear Area–Storage Equation

Following examples in the literature, e.g., the GRanD database, the intrinsic nonlinear area–storage relationship was formulated using a power function of form

$$A_t = a\left(S'_t\right)^b \tag{4}$$

where $A_t$ is the reservoir surface area ($10^6$ m$^2$) at time t, $S'_t$ is the corresponding gross (active + dead) storage volume ($10^6$ m$^3$) at $t$, and $a$ and $b$ are the parameters of the model. The parameters of Equation (4) were obtained by least-squares regression fitting using the available area–storage data.

#### 2.3.2. Single Linear Area–Storage Equation

Although the area–storage function is intrinsically nonlinear. as illustrated in Figure 1a, the commonly adopted approach is to approximate the entire part of the function above the dead storage using a linear function of the form [8,9]:

$$A_t = c + d(S_t); \; 0 \leq S_t \leq K_a \tag{5}$$

where $c$ and $d$ are parameters, $S_t$ is the active storage state, $K_a$ is the active storage capacity of the reservoir, and all other variables are as defined previously. In particular, parameter "$c$" is constrained to be the exposed surface area at the top of the dead storage; thus, "$d$" is the slope of the linear approximation in the active storage part. If "$c$" is constrained to $K_d$, the dead storage, the only parameter needing estimation in Equation (5) is slope "$d$". The linear approximation of the reservoir area–storage relationship is also shown schematically in Figure 1a.

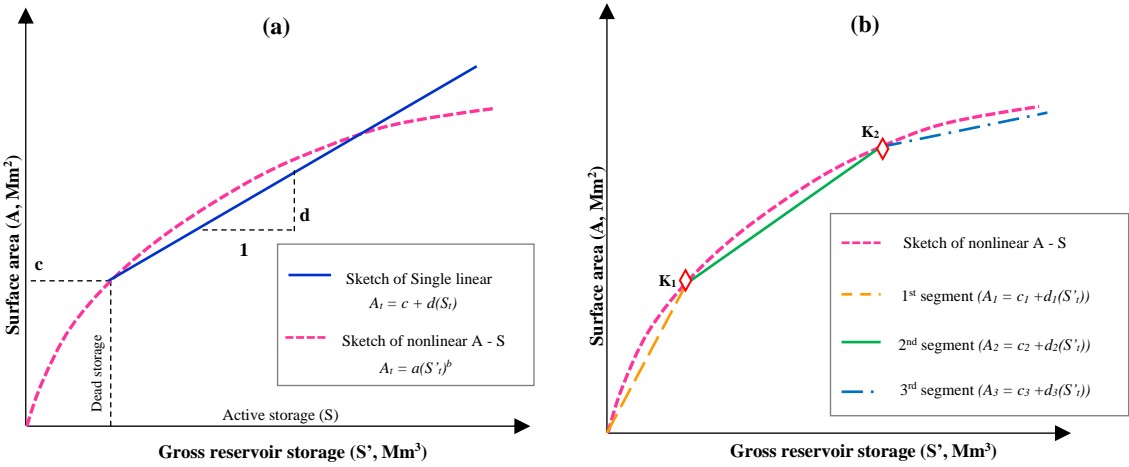

**Figure 1.** Sketch illustration of (**a**) single linear and (**b**) 3-piecewise linear approximations.

#### 2.3.3. Multiple (3) Piecewise Linear Area–Storage Equations

The use of a single linear function for the entire active storage zone can produce significant errors in the predicted area depending on the convexity of the function. A way to resolve this problem is to have multiple, piecewise linear functions. Specifically, for the current study, 3 piecewise linear area–storage functions (see Figure 1b) were considered:

$$A_1 = c_1 + d_1\left(S'_t\right); \; if \; S'_t < K_1 \tag{6}$$

$$A_2 = c_2 + d_2(S_t'); \; if \; K_1 \leq S_t' < K_2 \tag{7}$$

$$A_3 = c_3 + d_3(S_t'); \; if \; S_t' \geq K_2 \tag{8}$$

where, $K_1$ and $K_2$ are, respectively, the break points for the first and second models, and all other variables are as defined previously.

Thus, at storage $K_1$, the estimates of Equations (6) and (7) coincide, i.e.,

$$c_1 + d_1 K_1 = c_2 + d_2 K_1 \tag{9}$$

Rearranging Equation (9) yields:

$$c_2 = c_1 + K_1(d_1 - d_2) \tag{10}$$

Similarly, at $K_2$, the estimates of Equations (7) and (8) coincide, i.e.,

$$c_2 + d_2 K_2 = c_3 + d_3 K_2 \tag{11}$$

leading to

$$c_3 = c_2 + K_2(d_2 - d_3) \tag{12}$$

Putting the expression for $c_2$ from Equation (10) into Equation (11) gives:

$$c_3 = c_1 + K_1(d_1 - d_2) + K_2(d_2 - d_3) \tag{13}$$

Substituting Equation (10) in Equation (7), and Equation (12) in Equation (8) gives revised forms of the piecewise Equations for $A_2$ and $A_3$ as:

$$A_2 = c_1 + K_1(d_1 - d_2) + d_2 S_t'; \; if \; K_1 \leq S_t' < K_2 \tag{14}$$

$$A_3 = c_1 + K_1(d_1 - d_2) + K_2(d_2 - d_3) + d_3 S_t'; \; if \; S_t' \geq K_2 \tag{15}$$

The expression for $A_1$ in Equation (6) remains unchanged. Equations (6), (14) and (15) constitute a set of equations whose solution produces the parameters of the models, i.e., slopes ($d_1$, $d_2$ and $d_3$) as well as the lower limits ($c_1$, $K_1$ and $K_2$) for each segment of the continuous, piecewise linear functions.

In this study, a constrained optimization problem was formulated whose objective (fitness) function was the minimization of the sum of the squares of the residuals of the estimated reservoir surface area. The constrained optimization was solved using genetic algorithms to determine the parameters.

### 2.3.4. Nonlinear Height–Storage Equation

As noted earlier, the WEAP approach uses the height–storage relationship to account for evaporation. Although implementation in WEAP uses interpolation from the available height–storage data, it was felt that having a functional relationship makes the associated analysis simpler and more exact. Thus, in a manner similar to the nonlinear area–storage relationship, a nonlinear function for height–storage was formulated as:

$$H_t = h(S_t')^n \tag{16}$$

where $h$ and $n$ are parameters and all the other symbols are as previously defined. The parameters of Equation (16) were also determined by least-squares regression fitting using the available height–storage data.

### 2.3.5. GRanD Volume–Area Equation

Given the popularity with which the GRanD models are used for regional water-resource assessment across the globe [16], it was felt that this study would not be complete without testing

its efficacy at the case-study reservoirs. The generalised GRanD volume–area regression model was developed from the Global Reservoir and Dam database using 5824 reservoirs [11]. Its main purpose is to estimate missing reservoir volumes around the world. Although both volume–area and volume–height–area equations are available in the GRanD database, this study only deployed the volume–area equation to ensure consistency with the other models in the research.

The GRanD volume–area model takes the particular form:

$$S_t' = 30.68(A_t)^{0.978} \tag{17}$$

where $A_t$ is the reservoir surface area at time $t$ ($10^6$ m$^2$ or km$^2$); $S_t'$ is the corresponding reservoir total storage volume ($10^6$ m$^3$). It should be noted that the scale and exponent in Equation (17) are fixed as universal.

As noted earlier, GRanD analysis was principally concerned with predicting reservoir capacity (i.e., maximum storage) from the corresponding exposed surface area, and the data used for its calibration involve storage capacity and the corresponding maximum exposed surface area of the candidate reservoirs included in the analysis. Consequently, it is not a tool for describing how the exposed area varies with increasing stored volume behind a dam. It should therefore not be surprising if the function performs poorly when used for the purpose of predicting the at-site area–storage relationship. For example, while Lehner et al. [11] recorded an $R^2$ of 0.8 during the development of Equation (17), the use of the equation to predict the at-site area–storage relationship by van Bemmelen et al. [16] only produced an $R^2$ of 0.54; however, most analysts still use the GRanD equation for the latter purpose. Indeed, the GRanD model is used for this same purpose in the current study to further demonstrate its poor performance in local at-site situations.

### 2.4. Performance Assessment of H–A–S Formulations

The relative performance of various H–A–S functions was assessed using the $R^2$ metric:

$$R^2 = \left[ \frac{\sum_{i=1}^{n}\left(A_{b,i} - \mu_b\right)^2 - \sum_{i=1}^{n}\left(A_{b,i} - A_{m,i}\right)^2}{\sum_{i=1}^{n}\left(A_{b,i} - \mu_b\right)^2} \right] \tag{18}$$

where $A_{b,i}$ is the $i$th observed variable (area or height), $A_{m,i}$ is the $i$th variable as predicted by the model, and $\mu_b$ is the mean of the observed variable.

## 3. Results and Discussion

### 3.1. Case Study

As case studies, the methodology was tested on two reservoir systems in northern India: the Pong reservoir and the Bhakra reservoir as shown in Figure 2, which also shows the Pandoh dam, whose primary purpose is to divert some Beas River flows into the Bhakra reservoir.

The Pong reservoir (also Maharana Pratap Sagar) is a multipurpose (irrigation and hydropower generation) reservoir system constructed on the River Beas, India [17]. Constructed in 1975 of earth fill, the reservoir is located at latitude 31°59′02″ North and longitude 76°03′12″ East. The reservoir is located at the Himachal Pradesh District of Kangra at the valley of the Himalayas of the Gangetic plains [17]. This district experiences two seasonal regimes known as the monsoon (rainy) and post monsoon (dry) seasons. The former occurs from June to September, and the latter spans from October to May each year. The reservoir's catchment receives a mean annual precipitation of about 1800 mm [17]; the mean of the annual net evaporation is 493 mm.

The Bhakra reservoir (known locally as the Govind Saga Dam) is also a multipurpose reservoir system serving flood control, irrigation, and hydroelectric-power generation functions [17]. The reservoir was constructed on the River Satluj, in the Himachal Pradesh District of Bilaspur. It lies at

latitude 31°24′39″ North and longitude 76°26′02″ East. Just like the Pong reservoir in the Himalayas valley, the district experiences two seasonal regimes, monsoon (rainy) and post monsoon (dry). Mean annual rainfall for the catchment is approximately 1260 mm [16]; mean annual net evaporation is 571 mm.

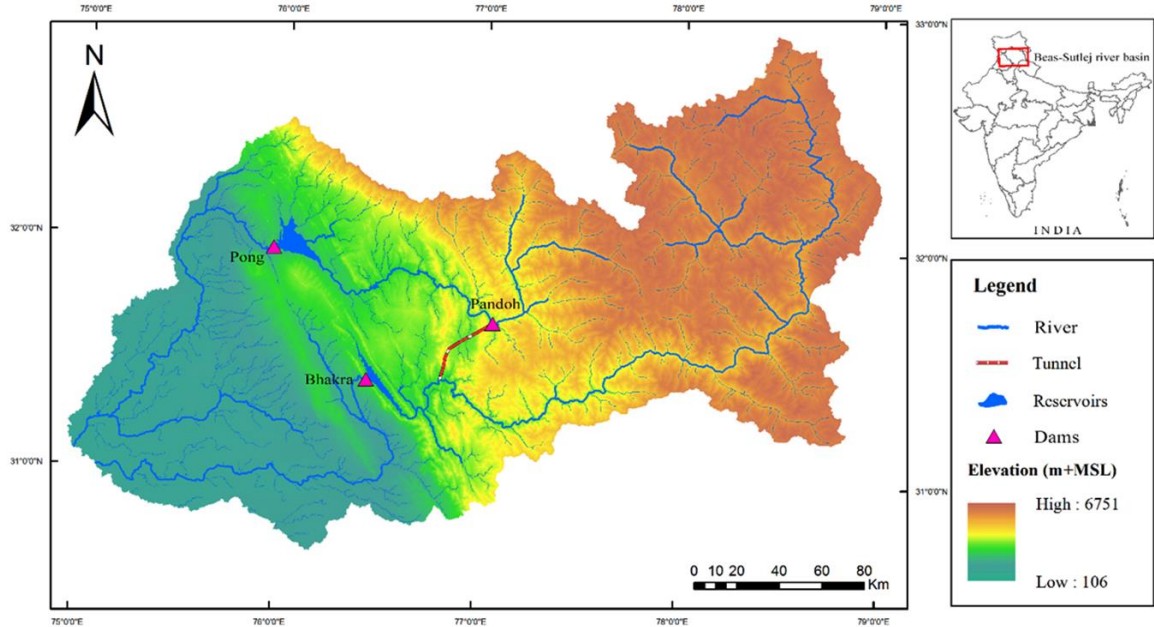

**Figure 2.** Locations of the main Pong and Bhakra reservoirs and the Pandoh dam diversion.

Further particulars about both reservoirs are available in Table 1. An interesting feature in Table 1 is that, while storage capacity at Bhakra exceeds that at Pong, the exposed surface area at full capacity at the latter is larger. This would suggest that the terrain at the Pong reservoir is plainer, making the potential volumetric evaporation to be more at a given storage state.

**Table 1.** Main design characteristics of the Pong and Bhakra reservoirs [18].

| Description | Pong Reservoir | Bhakra Reservoir |
|:---:|:---:|:---:|
| Catchment area (km$^2$) | 12,560 | 56,980 |
| Surface area at full capacity (km$^2$) | 240 | 162.48 |
| Gross storage capacity (Mm$^3$) | 8570 | 9621 |
| Active (live) storage capacity (Mm$^3$) | 7290 | 7191 |
| Dead storage capacity (Mm$^3$) | 1280 | 2430 |
| Elevation at top of dam (masl.) | 435.86 | 518.16 |
| Height above river bed (m) | 61 | 167.64 |
| Minimum annual flow (Mm$^3$) | 5211 | 12,346 |
| Maximum annual flow (Mm$^3$) | 9621 | 18,928 |
| Mean annual flow (Mm$^3$) | 7621 | 16,567 |
| CV | 0.20 | 0.15 |

The main inflows into both reservoirs derive from the runoff of monsoon rainfall; however, a sizeable contribution comes from the melting of glaciers and seasonal snow. Additionally, the Bhakra receives diverted water from the Beas at the Pandoh dam to augment its hydropower potential. The summary statistics of the annual inflows at the two reservoirs are also reproduced in Table 1; the Bhakra inflows include the Pandoh dam diversion. In general, interannual variability of the inflows at both sites is low (CV ≤ 0.2), signifying the dominance of within-year storage requirements in comparison to over-year requirements, which should be expected given the distinct seasonality of the inflows.

Annual net evaporation values are positive, implying that, on an annual basis, evaporation exceeds rainfall. As noted by Nawaz et al. [19], failure to accommodate net evaporation in planning

analysis for such a situation leads to undersizing of a reservoir because positive net evaporation is an additional demand that must be provided for. The monthly behavior of net evaporation is shown in Figure 3, which is a mixture of positive and negative values, as expected.

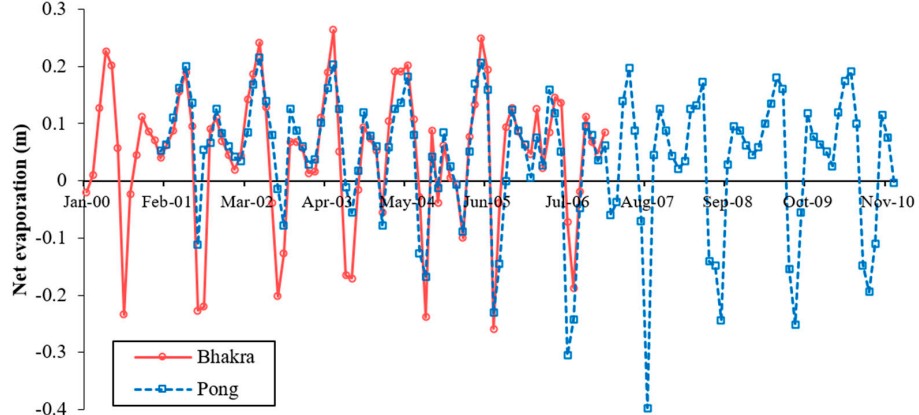

**Figure 3.** Net monthly depth of evaporation losses (m) at the Pong and Bhakra reservoirs.

### 3.2. Height–Area–Storage Curves of the Reservoirs

The available topographic data for the two reservoir sites are plotted in Figure 4a (Storage–Area) and Figure 4b (Storage–Height).

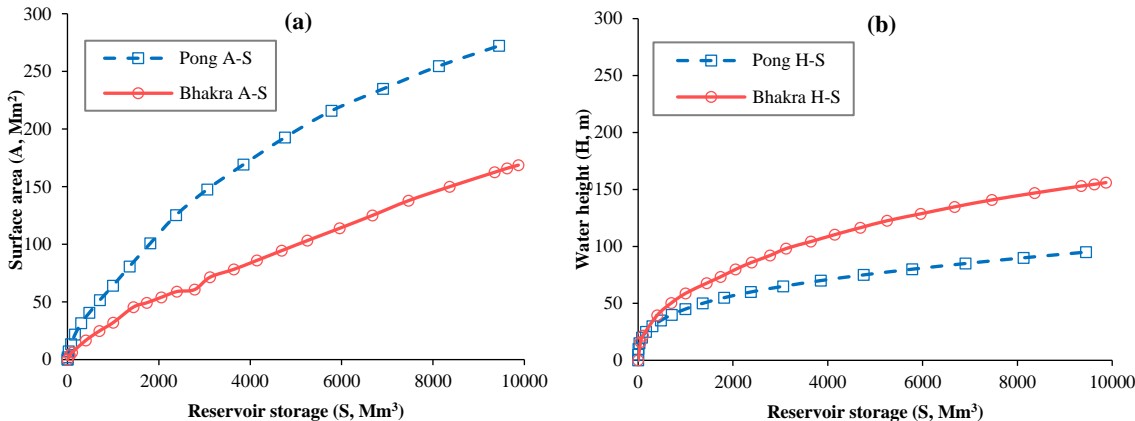

**Figure 4.** Plotted available topographical data for (**a**) Area–Storage and (**b**) Height–Storage at Pong and Bhakra Reservoirs.

The steepness of the slopes for each relationship has an influence on the rate of evaporation loss from the reservoir. As seen in Figure 4a, the area–storage relationship for Pong is steeper than Bhakra, implying that, at the former site, a small change in reservoir storage significantly alters the exposed surface area and, by extension, the evaporation loss from the reservoir. This would mean that the earlier observation regarding the relative spread of both reservoirs, as presented in Table 1, is not restricted to the top water level alone, but applies throughout the entire area–storage range at the Pong and Bhakra reservoir sites.

### 3.3. Height–Area–Storage Models of the Reservoirs

The calibrated H–A–S models are presented in Table 2 for both reservoirs. As a reminder, all empirical functions apart from the GRanD were calibrated using H–A–S data for the respective reservoir sites. The GRanD model was taken directly from the database and is therefore the same for both reservoirs.

**Table 2.** Calibrated Pong and Bhakra H–A–S models.

| Models | Pong Reservoir | | | Bhakra Reservoir | | |
|---|---|---|---|---|---|---|
| | **Equation** | **RMSE** | **$R^2$** | **Equation** | **RMSE** | **$R^2$** |
| Single Linear A–S | $A = 77.17 + 0.0311(S)$ | 15.13 | 0.960 | $A = 59.16 + 0.0152(S)$ | 1.92 | 0.997 |
| 3-Piecewise Linear A–S | $A_1 = 5.91 + 0.0613(S')$ *(if S' < 1280 Mm³)* $A_2 = 40.68 + 0.0341(S')$ *(if 1280 ≤ S' < 4365 Mm³)* $A_3 = 116.2 + 0.0168(S')$ *(if S' > 4365 Mm³)* | 4.18 | 0.970 | $A_1 = 17.18 + 0.0185(S')$ *(if S' < 2430 Mm³)* $A_2 = 20.32 + 0.0172(S')$ *(if 2430 ≤ S' < 7276 Mm³)* $A_3 = 32.23 + 0.0155S'$ *(if S' > 7276 Mm³)* | 3.52 | 0.970 |
| Nonlinear A–S | $A = 0.7773538(S')^{0.6492}$ | 3.03 | 0.998 | $A = 0.2284(S')^{0.7158}$ | 2.77 | 0.993 |
| Nonlinear H–S | $H = 4.851627(S')^{0.3234}$ | 0.54 | 0.999 | $H = 2.9344(S')^{0.4339}$ | 1.42 | 0.997 |
| GRanD Nonlinear S–A | $S' = 30.684(A)^{0.9578}$ | 22.72 | 0.800 | $S' = 30.684(A)^{0.9578}$ | 133.24 | 0.800 |

Note: S and $S'$, active and total (active + dead) storage reservoir states, respectively; H–A–S, Height–Area–Storage; RMSE, root mean square error.

As the $R^2$ values indicate, nonlinear functions are better than linear ones, which should be expected from the intrinsically nonlinear nature of the relationship. Using three piecewise linear approximations for the area–storage relationship appears to improve the prediction. As found in the previous independent studies discussed earlier, the performance of the GRanD model is by far inferior to all empirical models calibrated in the current study.

The predicted and observed area–storage relationships are compared in Figure 5a,b for the Pong and Bhakra reservoirs, respectively. As a reminder, the single linear model only applies above the dead storage, while the other models also cover the dead storage zone. These further confirm the superiority of the nonlinear approximation over the single, linear approximation. Indeed, for much of the active storage-capacity regions at both reservoirs, the single linear approximation underpredicts the exposed surface area of the reservoir, although the situation was more marked at the Pong reservoir, with maximum underprediction being about 11.2%. Since planning analysis is primarily concerned with estimating reservoir active storage capacity, it means one could expect this reservoir capacity to be undersized with linear approximation if net evaporation is positive. The behavior of the linear model changes to overprediction of the exposed area at very high storage capacity, reaching a maximum overprediction error of 21%, although as shown in Figure 5a, overprediction only occurred for a very narrow range of storages.

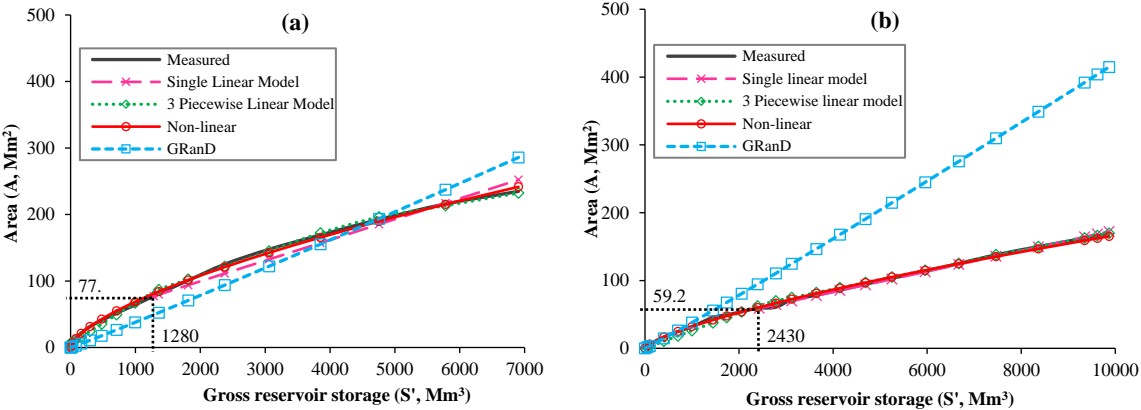

**Figure 5.** Available and predicted surface areas for the (**a**) Pong and (**b**) Bhakra reservoirs.

The performance of the GRanD model is not the same for the two reservoirs. While as seen in Figure 5a, the GRanD model largely underestimates the exposed area of the reservoir at the Pong, the opposite is true for the Bhakra. For the Pong reservoir, where the bias is downward, bias size exceeds the single linear approximation, reaching over 100%; consequently, the capacity undersizing in situations of positive net evaporation is expected to be accentuated when the GRanD model is applied. The upward bias of the GRanD model for the Bhakra reservoir is even worse, reaching a maximum of about 415%. Although an overestimate error of 370% was previously recorded by van Bemmelen et al. [16] with the GRanD model, the maximum error of 415% recorded in this study for the Bhakra reservoir is further proof that the GRanD model is an unreliable tool for at-site area–storage predictions. Several regional models have been compared with the GRanD equation, and it generated poor results [20–22]. Because it overestimates the exposed area of reservoirs, its use results in oversizing of the reservoir capacity in situations where net evaporation is positive, but undersizing it when net evaporation is negative.

The performance of the fitted nonlinear height–storage relationships is shown in Figure 6a,b, which is in line with the high $R^2$ values reported earlier. The fact that the fitted nonlinear height–storage relationship performs better than its area–storage counterpart has been noted elsewhere [2,23–25]. It is therefore not surprising that the height–storage function has been preferred in implementing evaporation consideration in the WEAP tool.

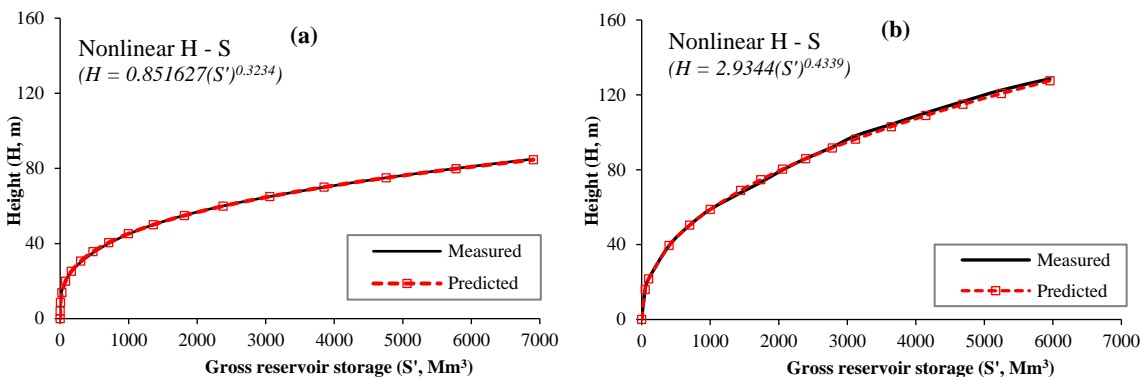

**Figure 6.** Available and predicted mean nonlinear H–S for the (**a**) Pong and (**b**) Bhakra reservoirs.

### 3.4. Assessed Effects of Evaporation Loss on Reservoir Storage Capacity

Reservoir capacity estimates are presented in Tables 3 and 4 for the Pong and Bhakra reservoirs, respectively. These are for various assumed demands (or yields), expressed as fraction of the mean annual runoff (MAR) at the respective sites. Reservoir planning analysis was performed using a monthly data record and, given the marked seasonality of the rainfall in the two catchments as noted previously, net evaporation is a mixed bag of positive and negative values, with the majority of the positive values occurring during the postmonsoon months (see Figure 4). It is therefore very unlikely that the expected impact of the net evaporation would be as high as one would expect if all monthly net evaporation values were positive. This might change if the projected decrease in monsoon rainfall, coupled with rises in both temperature and evaporation as a result of climate change, take hold. However, such issues are not within the purview of the current study and have therefore not been considered.

**Table 3.** Pong reservoir active storage-capacity estimates ($10^6$ m$^3$).

| Yield (MAR) | Without Evaporation | With Evaporation for Different H–A–S Formulations | | | | |
|:---:|:---:|:---:|:---:|:---:|:---:|:---:|
| | | Single Linear | Multiple Linear | Nonlinear A–S | Nonlinear H–S | GRanD A–S |
| 0.2 | 11.5 | 13.3 | 13.1 | 13.1 | 13 | 12.5 |
| 0.4 | 76.9 | 81.8 | 82.4 | 82.3 | 82.1 | 80.3 |
| 0.6 | 167.6 | 175.6 | 176.4 | 176.3 | 175.9 | 174.1 |
| 0.7 | 225.6 | 239.2 | 240.5 | 240.1 | 239.1 | 237.4 |
| 0.8 | 336.8 | 352.5 | 353.7 | 353.4 | 352.5 | 351.5 |
| 0.9 | 447.9 | 465.9 | 466.6 | 466.4 | 465.6 | 465.6 |
| 0.98 | 510.7 | 529.4 | 529.5 | 529.7 | 524.8 | 529.2 |

Note: MAR, mean annual runoff; GRanD, Global Reservoir and Dam.

**Table 4.** Bhakra reservoir active storage-capacity estimates ($10^6$ m$^3$).

| Yield (MAR) | Without Evaporation | With Evaporation for Different H–A–S Formulations | | | | |
|:---:|:---:|:---:|:---:|:---:|:---:|:---:|
| | | Single Linear | Multiple Linear | Nonlinear A–S | Nonlinear H–S | GRanD A–S |
| 0.2 | 2.8 | 3.6 | 3.6 | 3.6 | 3.4 | 4.1 |
| 0.4 | 82.8 | 85.2 | 85.4 | 85.3 | 84.6 | 86.9 |
| 0.6 | 219.5 | 222.8 | 223.1 | 223.0 | 222.1 | 226.1 |
| 0.7 | 309.0 | 313.7 | 313.9 | 313.8 | 312.7 | 318.4 |
| 0.8 | 400.2 | 405.3 | 405.5 | 405.4 | 404.1 | 410.6 |
| 0.9 | 585.5 | 593.5 | 593.7 | 593.6 | 591.6 | 602.5 |
| 0.98 | 758.7 | 768.3 | 768.4 | 768.2 | 766.0 | 780.3 |

Nonetheless, as seen in Tables 3 and 4, consideration of evaporation has resulted in higher reservoir capacity estimates compared to when evaporation was ignored. In general, for both reservoirs, the additional capacity required to cope with evaporation increases as demand increases, although in proportional terms, the reverse is the case due to increasing capacity with demand. The proportional requirement for the 0.2 MAR demand was very high due to the small capacity estimate for this demand.

Given the good performance of the nonlinear height–storage function (see Figure 6a,b), one would expect its estimate of reservoir capacity to be the best, and should therefore form the benchmark for the assessment of the other H–A–S models. For the Pong reservoir (see Table 3), the GRanD model produced the least evaporation requirement because of its negative bias in the prediction of the exposed reservoir surface. The single linear approximation model also produced a downward bias in exposed area predictions, but this bias was less than that of GRanD and has translated into a higher evaporation requirement when compared to that of GRanD. The nonlinear area–storage model produced the closest to the nonlinear height–storage in terms of the evaporation requirement, which should be expected from the performance of the nonlinear area–storage model in predicting the exposed reservoir surface area.

The results in Table 4 for the Bhakra reservoir convey the same outcome as that of the Pong, although evaporation requirements in proportional terms were much lower for the former. Another feature of the results in Table 4 is the higher capacity estimate for the GRanD when compared to the nonlinear height–storage model. As noted previously, the GRanD model, unlike its performance at the Pong reservoir, produced an overprediction of the exposed surface area at Bhakra, which has translated into higher reservoir capacity when compared to the benchmark height–storage model.

The required corrections for evaporation in capacity estimates are graphed in Figure 7a,b to better illustrate how these change with the adopted H–A–S model. As shown in the figures, the evaporation adjustment is indistinguishable for all H–A–S models, the only exception being the GRanD model, whose adjustments were marginally different. Fennessey [26] investigated the impact of evaporation data time step on capacity adjustment during planning analysis and concluded that using time series evaporation data produced almost indistinguishable results from using lumped, seasonal average values of evaporation. There are two possible reasons for the present outcome and that of Fennessey. First is that, as remarked earlier, the alternating positive and negative values of monthly net evaporation could be responsible for subduing evaporation-loss correction. However, a second and perhaps much more important reason concerns the size of the evaporative demand relative to the tangible consumptive demand. For example, total annual evaporative demand for the Pong reservoir, assuming a constant surface area of 240 km$^2$, is only 118 Mm$^3$, which, compared to the consumptive demand at, e.g., 0.2 MAR (=762.1 Mm$^3$), is a mere 7% of the total demand. As consumptive demand grows, the contribution of evaporative demand to the total annual demand becomes even more insignificant, dropping to below 2% at 0.8 MAR.

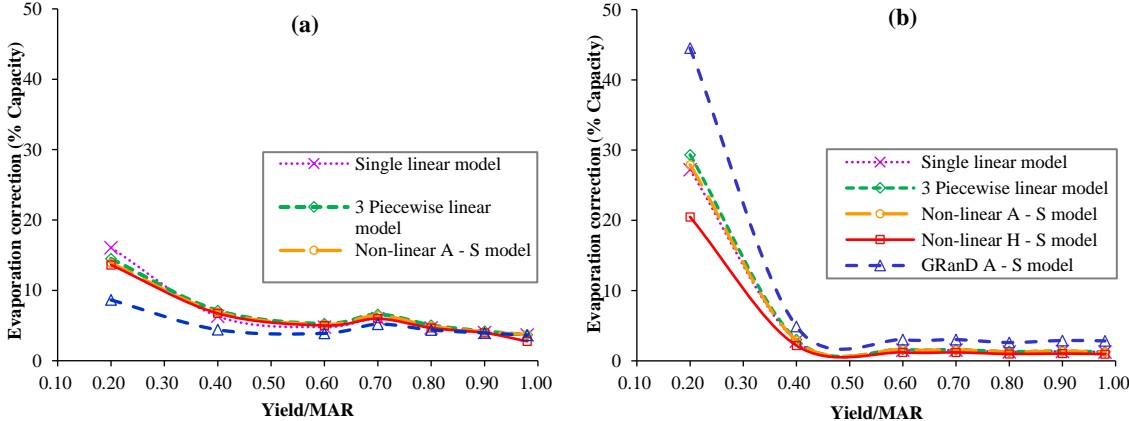

**Figure 7.** Evaporation correction as % of capacity in (**a**) Pong and (**b**) Bhakra reservoirs.

Although the H–A–S models are almost indistinguishable in terms of their capacity effects, there are discernible yield effects. For example, for the Pong reservoir, all the different H–A–S models are consistent in their trajectories by first declining until the demand of 0.6MAR, after which they rise slightly at 0.7MAR before resuming the declining trend. The same behavior was exhibited by the trajectories for the Bhakra, although as noted previously, the capacity needed to cater for evaporation in proportional terms is lower than that in Pong. Additionally, the rise at 0.7 MAR was less noticeable at Bhakra, but could still be discerned. The reason for the slight rise at demand of 0.7 MAR is not immediately clear, but has also been previously observed by Montaseri and Adeloye [7] who analyzed data catchments in Iran and the UK.

Given the proven superiority in this study of the height–storage model for incorporating evaporation in reservoir-planning analysis, Figure 8 was produced by averaging its results at both Pong and Bhakra that can be used as a tool for evaporation correction within the Beas–Sutlaj complex in the Indus. Figure 8 can therefore be thought of as a regional tool for evaporation correction during reservoir planning. To use it for this purpose, analysis is first carried out without consideration of evaporation using a suitable planning technique, e.g., the basic SPA where the site is gauged, or one of the indirect techniques [8,27] for ungauged sites. From the known demand/MAR to be met by the reservoir, Figure 8 is entered, and the required evaporation correction is read off from the plot. The adjusted reservoir capacity estimate then becomes the sum of without-evaporation capacity estimate and evaporation correction.

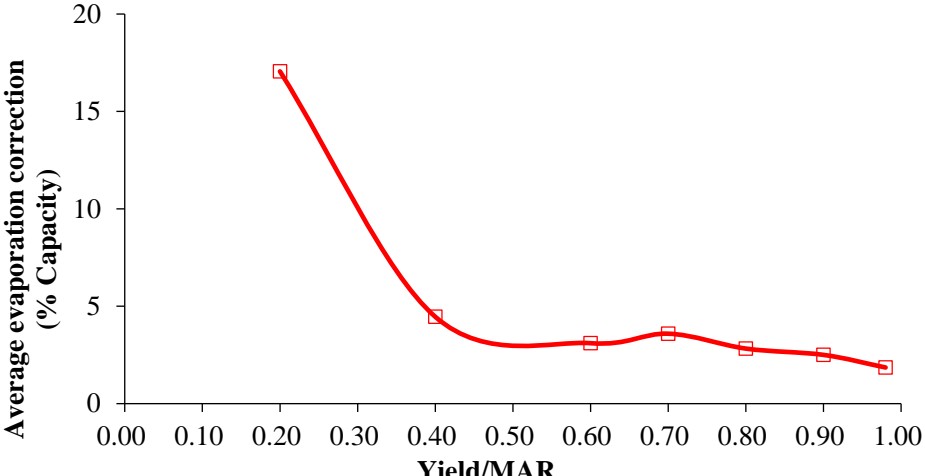

**Figure 8.** Evaporation correction as % of capacity (average for Pong and Bhakra reservoirs with the nonlinear H–S model).

Thus, this study has developed a tool that could serve the Indus Basin region of India for the purpose of correcting for evaporation loss during reservoir planning. While the tool could be considered to be of limited scope, being based on just two reservoir systems, it shows that the development of such a useful tool is possible and can be enhanced by more analyses of other case studies. This should encourage further studies in the region and, indeed, other regions of the world.

## 4. Conclusions

This study has demonstrated that planning analyses of surface-water reservoirs without the explicit accommodation of evaporation loss leads to errors in storage-capacity estimates. Specifically, analyses reported here showed that an increase of up to 29% in reservoir storage capacity is required to compensate for the effect of net evaporation loss, depending on the applied H–A–S model. The output of the research is thus expected to positively contribute to the practice and knowledge of reservoir-planning and operational analyses.

A further notable outcome of the study is the extremely poor performance of the GRanD H–A–S models in predicting at-site H–A–S, although its capacity estimates were comparable with those of the other H–A–S models. This finding with regard to bias in area is significant because of the increasing use of the GRanD H–A–S models by analysts, either to predict reservoir capacity from the known surface area or as input into reservoir-planning analysis to accommodate volumetric evaporation loss.

**Author Contributions:** Conceptualization, A.J.A.; Methodology, A.J.A., I.Y.W., and Q.V.D.; Validation, A.J.A., I.Y.W., and Q.V.D.; Formal analysis, A.J.A., and I.Y.W.; Investigation, A.J.A.; Resources, Q.V.D., B.-S.S., and K.S.K.; Writing—original-draft preparation, A.J.A., I.Y.W.; Writing—review and editing, A.J.A., I.Y.W., and Q.V.D.; Visualization, I.Y.W. and Q.V.D.; Supervision, A.J.A.; Funding acquisition, A.J.A.

**Funding:** This work reported here forms part of "Sustaining Himalaya Water Resources in a Changing Climate (SusHi-Wat)" funded by the UK-Natural Environment Research Council (UK-NERC) grant number NE/N016394/1 as part of the UK–India Newton–Bhabha Sustainable Water Resources (SWR) thematic Program.

**Conflicts of Interest:** The authors declare no conflict of interest.

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
