# Peer review of "Height–Area–Storage Functional Models for Evaporation-Loss Inclusion in Reservoir-Planning Analysis"

_water, doi:10.3390/w11071413_

Round 1
Reviewer 1 Report
First, this study seems important to me, given Nepal's/India's critical snowmelt/climate forcing problem and the hundreds of millions of people's survival directly and imminently affected.
Second, lines 89-91: "topographical data in the form of
corresponding H-A-S relationship at the analyzed reservoir sites will be required for developing the
empirical functions to incorporate the evaporation loss directly in the planning analysis."
contrast or seem incongruous with lines 96-99: "Additionally, as will
become clearer in 3.2, the contrasting shape revealed by the available topographical data will make
possible generalization of the outcome feasible and is thus another reason for choosing the two
reservoir systems."
So, authors, is this study fundamentally empirical or is it presenting generalizable & working theory (beyond upland India reservoirs)? To strongly assert the generalizability, present a model that's empirically calibrated and also validated with separate but comparable data. Validation could be from other Bhakra-Beas Management Board reservoirs(?) or from similar data in other countries, etc. Without strong validation using separate data, it's difficult for a reviewer to become convinced of the generalizability of a 2-sample model. However, this study focuses on a critical water need. Perhaps you could validate this work in a future publication.
Near line 194, the piecewise linear approximations set will always underestimate (i.e., the linear functions are always at or below the nonlinear function). The error is smaller than the single linear estimator function (graph a), but the error is now systematically in the direction of underestimation. Is this an acceptable scale of error?
Why not present and test a simple (few attributes) but detailed (many rows/entries) lookup table model of discrete measured gross reservoir storage vs. area? Any spreadsheet or GIS can make the lookup trivially fast, instead of having to solve the nonlinear model (also pretty fast with good computers/optimizations these days). The lookup table would represent a piecewise linear approximation where the number of pieces is arbitrarily large (a hundred, a thousand) segments and the model fitment error is correspondingly small (R-squared > 0.999).
At line 377, I would like to see (and a robust, generalizable model would provide) a quantitation of the risk statement: "It is therefore very unlikely that the expected impact of the net evaporation would be as high as one would expect if all the monthly net evaporation values were positive."
I believe your "very unlikely" but with ~500,000,000 to 600,000,000 Indians currently facing water exhaustion (http://www.niti.gov.in/writereaddata/files/document_publication/2018-05-18-Water-index-Report_vS6B.pdf), knowing that risk size can mean literal survival or death for some fraction of the country's population. Lives hang in the balance. You can strengthen your manuscript's assertions here.
Author Response
Manuscript title:
Height-Area-Storage Functional Models for Evaporation Loss inclusion in Reservoir Planning Analysis (water-492087)
We highly appreciate the comments from the reviewers, which have helped us in improving the quality of the manuscript. We have carefully addressed the comments and our response to the specific questions are presented in the table below.
Reviewer 1 | Author’s Responses |
Comment 1: lines 89-91: "topographical data in the form of corresponding H-A-S relationship at the analyzed reservoir sites will be required for developing the empirical functions to incorporate the evaporation loss directly in the planning analysis." contrast or seem incongruous with lines 96-99: "Additionally, as will become clearer in 3.2, the contrasting shape revealed by the available topographical data will make possible generalization of the outcome feasible and is thus another reason for choosing the two reservoir systems."
So, authors, is this study fundamentally empirical or is it presenting generalizable & working theory (beyond upland India reservoirs)?
To strongly assert the generalizability, present a model that's empirically calibrated and also validated with separate but comparable data. Validation could be from other Bhakra-Beas Management Board reservoirs(?) or from similar data in other countries, etc. Without strong validation using separate data, it's difficult for a reviewer to become convinced of the generalizability of a 2-sample model. However, this study focuses on a critical water need. Perhaps you could validate this work in a future publication. | We have removed the second sentence to avoid this apparent conflict.
Yes, we appreciate that independent validation is crucial for generalization and we would like to do this but the data for other reservoirs have been difficult to obtain. We are in discussion with the BBMB and as soon as we obtain the data, we will be embark on the independent validation of the results and will report the findings in a different publication.
|
Comment 2: Near line 194, the piecewise linear approximations set will always underestimate (i.e., the linear functions are always at or below the nonlinear function). The error is smaller than the single linear estimator function (graph a), but the error is now systematically in the direction of underestimation. Is this an acceptable scale of error? | The diagram in Figure 1 are for illustration only and the bias have therefore been exaggerated. However, looking at the calibrated linear (single and piece-wise linear) models in Figure 5, these biases are not very large and not consistently downward. |
Comment 3: Why not present and test a simple (few attributes) but detailed (many rows/entries) lookup table model of discrete measured gross reservoir storage vs. area? Any spreadsheet or GIS can make the lookup trivially fast, instead of having to solve the nonlinear model (also pretty fast with good computers/optimizations these days). The lookup table would represent a piecewise linear approximation where the number of pieces is arbitrarily large (a hundred, a thousand) segments and the model fitment error is correspondingly small (R-squared > 0.999). | Yes, interpolating from a look-up table is an option; indeed, that is the approach used in the WEAP model. However, we felt automating the analysis was better if functional relationships were used, which is why we developed the functions.
|
Comment 4: At line 377, I would like to see (and a robust, generalizable model would provide) a quantitation of the risk statement: "It is therefore very unlikely that the expected impact of the net evaporation would be as high as one would expect if all the monthly net evaporation values were positive."
I believe your "very unlikely" but with ~500,000,000 to 600,000,000 Indians currently facing water exhaustion (http://www.niti.gov.in/writereaddata/files/document_publication/2018-05-18-Water-index-Report_vS6B.pdf), knowing that risk size can mean literal survival or death for some fraction of the country's population. Lives hang in the balance. You can strengthen your manuscript's assertions here. | Thanks- some further comments have been appended to this statement to qualify the likelihood. |

Reviewer 2 Report
Height-Area-Storage Functional Models for Evaporation Loss inclusion in Reservoir Planning Analysis
General comments: I have carefully reviewed the article entitled “Height-Area-Storage Functional Models for Evaporation Loss inclusion in Reservoir Planning Analysis”. The article is within the scope of this journal and presents novel results on the evaluation of different models and their functions as well as biases resulting from using various models (H-A-S) in reservoirs.
This study makes it an interesting case study with global implications. I find the article very interesting case. However, I have the following points to mention and there some other suggestions/corrections given in the text file (Pdf file). Please use the Foxit PDF reader to know the suggestions/corrections.
There are some comments, suggestions, as well as some words, are cut by overlapping lines. It is suggested to either delete or replace those words as I feel they do not need to be mentioned.
Title: The title is okay and comprehensive in its formation.
Abstract: The abstract is more like the part of the article ‘objectives and looks devoid of any results and discussion part’. The closing lines are not required to be mentioned here. Instead, add some conclusions here. I suggest you add some numerical outcomes to show the comparison between the various types of models used and remove the information about the database, as it does not belong here. Therefore, the abstract requires immediate attention with respect to its contents and English Grammar. Further, it needs to be enriched with a little more results and add conclusions sections so that it is of more interest to the targeted readership.
Introduction:
The introduction is generally well-written and illustrates the background knowledge and existing gaps. However, in my view, it needs to be enriched with more citations from the highly cited published literature. Some specific comments can be found in the pdf file and some are mentioned here.
L 43-45: Please mention the name of the author rather than just putting the citation number. So, it will be mentioned something like de Araujo et al. [6]. Please follow this in the whole MS.
L 68-69: effect of what?????
L 81-82: This does not belong here. Please move it to its relevant part.
L 83-85: This is nothing new. Every article goes like this. Please delete this.
Materials and Methods
L 88: “This study will require…….” The study is already complete and sent for publication; therefore, using the future tense makes no sense. Please go through the whole MS and should write in the past tense.
L96-99: This does not belong here. It might go to the results and discussion section.
L 268-322: This is obviously part of the results and discussion section. Therefore, it should be moved there.
Results and Discussion
L 377: it should be post-monsoon instead of ex-monsoon.
L 380: Table 3 caption should be more like containing more information e.g. names of reservoirs alone may confuse the readers, therefore use full information i.e. Pong reservoirs, etc.
Table 3: One field looks missing in the second column. Refer to the attached pdf file.
Table 4. Refer to the above comment.
Table 3 & 4: Please provide the detailed form of acronyms used in the tables at the bottom of each.
L 383-384: This line very confusing and unfinished. Perhaps you are talking about some comparison but you left it half-way. It does not yield any meanings.
Line 383-405: The lines are bit mixed and confusing. Please revise here carefully to yield reasonable meanings.
L 405-406: Figure 7 (a,b), please follow this throughout the MS.
L 407-408: Please rephrase to yield the meaning.
L 408: Please mention the name of the author in this citation.
L 412: please refer to the above comment.
Conclusions:
Please make one or maximum two paragraphs in this section.
Please shorten the conclusions section as it contains more information that is generic.
References:
Please look into the in-text and references list for format errors.

Author Response
Manuscript title:
Height-Area-Storage Functional Models for Evaporation Loss inclusion in Reservoir Planning Analysis (water-492087)
We highly appreciate the comments from the reviewers, which have helped us in improving the quality of the manuscript. We have carefully addressed the comments and our response to the specific questions are presented in the table below.
Reviewer 2 | Author’s Responses |
Comment 1: The abstract is more like the part of the article ‘objectives and looks devoid of any results and discussion part’. The closing lines are not required to be mentioned here. Instead, add some conclusions here. I suggest you add some numerical outcomes to show the comparison between the various types of models used and remove the information about the database, as it does not belong here. Therefore, the abstract requires immediate attention with respect to its contents and English Grammar. Further, it needs to be enriched with a little more results and add conclusions sections so that it is of more interest to the targeted readership. | We have changed the abstract and included the major results. |
Comment 2: L 43-45: Please mention the name of the author rather than just putting the citation number. So, it will be mentioned something like de Araujo et al. [6]. Please follow this in the whole MS. | Done! |
Comment 3: L 68-69: effect of what????? | The effects of net-evaporation loss on storage capacity estimates. The sentence has been re-written to make this clearer |
Comment 4: L 81-82: This does not belong here. Please move it to its relevant part. | Done! |
Comment 5: L 83-85: This is nothing new. Every article goes like this. Please delete this. | Done! |
Comment 6: L 88: “This study will require…….” The study is already complete and sent for publication; therefore, using the future tense makes no sense. Please go through the whole MS and should write in the past tense. | Done! |
Comment 7: L96-99: This does not belong here. It might go to the results and discussion section. | The sentence has been moved to Conclusion as suggested. |
Comment 8: L 268-322: This is obviously part of the results and discussion section. Therefore, it should be moved there. | Moved as suggested. |
Comment 9: L 377: it should be post-monsoon instead of ex-monsoon | Done! |
Comment 10: L 380: Table 3 caption should be more like containing more information e.g. names of reservoirs alone may confuse the readers, therefore use full information i.e. Pong reservoirs, etc. | The captions have been re-written as suggested. |
Comment 11: Table 3: One field looks missing in the second column. Refer to the attached pdf file. | Table’s format corrected. |
Comment 12: Table 4. Refer to the above comment. | Table’s format corrected. |
Comment 13: Table 3 & 4: Please provide the detailed form of acronyms used in the tables at the bottom of each. | Done! |
Comment 14: L 383-384: This line very confusing and unfinished. Perhaps you are talking about some comparison but you left it half-way. It does not yield any meanings. | This has now been re-written. |
Comment 15: Line 383-405: The lines are bit mixed and confusing. Please revise here carefully to yield reasonable meanings. | This has been done. |
Comment 16: L 405-406: Figure 7 (a,b), please follow this throughout the MS. | Done! |
Comment 17: L 407-408: Please rephrase to yield the meaning. | Done! |
Comment 18: L 408: Please mention the name of the author in this citation. | Done! |
Comment 19: L 412: please refer to the above comment. | Done! |
Comment 20: Please make one or maximum two paragraphs in this section. Please shorten the conclusions section as it contains more information that is generic. | The conclusion now has two paragraphs as suggested. |
Comment 21: Please look into the in-text and references list for format errors. | Done! |
Comment P1-1: No need of this in the abstract rather it shall go to the objectives section. | This sentence was deleted. |
Comment P1-2: Please enrich the abstract with more of the numerical outcomes based on your findings. | Please refer to the response for Comment #1. |
Comment P1-3: would be based on an wrong or biased measurement. | Done! |
Comment P1-4: Please mention the name of the author rather than just putting the citation number. | Please refer to the response for Comment #2. |
Comment P2-1: the evaporation loss in question explicitly........ | Done! |
Comment P2-2: persisting | Done! |
Comment P2-3: Effect of what? | Please refer to the response for Comment #3. |
Comment P2-4: Delete | Done! |
Comment P2-5: Move from here and might go to the conclusions section. | Please refer to the response for Comment #4. |
Comment P2-6: Delete this all. | Please refer to the response for Comment #5. |
Comment P3-1: Please mention the research program name as well. | Name of the research program was added in the revised manuscript. |
Comment P3-2: This does not belong here. | Please refer to the response for Comment #7. |
Comment P9-1: Which site are you referring to? | Pong and Bhakra reservoir sites. This has been updated in the revised manuscript. |
Comment P10-1: Please mention the name followed by the citation number. | Done! |
Comment P11-1: post-monsoon instead of ex-monsoon | Please refer to the response for Comment #9. |
Comment P11-2: Put the column name. | Please refer to the response for Comment #11. |
Comment P12-1: Without what? | Please refer to the response for Comment #14. |
Comment P12-2: This can not be the feature of a table but related to some model or the parameters used in this study. | Thanks- we have re-written this. |
Comment P12-3: Figure 7 (a,b), please follow this throughout the MS. | Done! |
Comment P12-4: Please rephrase. | Done! |
